# Temperature Dependence of Rubber Hyper-Elasticity Based on Different Constitutive Models and Their Prediction Ability

**DOI:** 10.3390/polym14173521

**Published:** 2022-08-27

**Authors:** Xiulong Yao, Zepeng Wang, Lianxiang Ma, Zhanli Miao, Minglong Su, Xiaoying Han, Jian Yang

**Affiliations:** College of Electromechanical Engineering, Qingdao University of Science and Technology, Qingdao 266061, China

**Keywords:** constitutive model, carbon black, filled rubber, finite element analysis (FEA)

## Abstract

Based on the electronic universal testing machine with a temperature chamber, this paper investigated the temperature and filler effects on the hyper-elastic behavior of reinforced rubbers and revealed the regulation of the stress and strain of the natural rubber and filled rubber with temperature. The experimental results showed that the hyper-elastic behavior of the filled rubber was temperature-dependent in a wide range. Comparing the adaptability of different models to the stress–strain variation with temperature, the Yeoh model was proven to reasonably characterize the experimental data at different temperatures. Based on the Yeoh model, an explicit temperature-dependent constitutive model was developed to describe the stress–strain response of the filled rubber in a relatively large temperature range. The prediction data of this proposed constitutive model fit well with the test data of the mechanical experiments, indicating that the model is suitable to characterize the large deformation behavior of filled rubbers at different temperatures to a certain degree. The proposed model can be used to obtain the material parameters and has been successfully applied to finite element analysis (FEA), suggesting a high application value. Notably, the model has a simple form and can be conveniently applied in related performance tests of actual production or finite element analysis.

## 1. Introduction

Rubber elastomers composed of long chains, macromolecules, and mesh-crosslinked structures have been commonly used in the automotive, aeronautical, and electronic industries. As a commonly used reinforcing agent for rubber products, carbon black has endowed natural rubber with better mechanical properties and thermo-elasticity. For carbon black-filled rubbers, temperature has a great influence on its hyper-elastic behavior. Due to the wide range of temperatures in a variety of applications [1,2], it is of great necessity to consider the impact of temperature on the hyper-elastic properties of carbon black-filled rubbers.

Filled rubber is usually used as damping and shock-absorbing components in the automotive and aerospace industries. Since the self-healing nature of elastic components in high-temperature environments or which are subjected to cyclic loading makes the temperature rise, the mechanical response of elastic components will be severely affected due to thermal coupling. [3] Although the mechanical responses of filled and unfilled rubber have been characterized at room temperature and high temperatures [4,5,6,7,8], the effects of temperature on the mechanical response of rubber materials in a certain deformation range, such as a 150% strain, have rarely been studied. From the point of view of tire applications and industrial formulations, it is necessary to carry out tests on the hyper-elastic mechanical properties of carbon black-filled rubber specimens at different temperatures in a wide range of deformations (150% strain) [9]. Meanwhile, both the filler-rubber matrix and filler–filler interactions at different temperatures also have a significant effect on the thermal behavior of rubber components [10].

The hyper-elastic mechanical behavior of rubber materials depends not only on the temperature but also the filler content. It is necessary to develop a temperature-dependent model to predict the hyper-elastic behavior of rubber based on the existing hyper-elastic constitutive model. The model is able to clearly describe and expose the temperature characteristics of rubber components with different carbon black contents [11]. Several thermodynamic models have been proposed to evaluate the effect of temperature on the mechanical properties of filled rubber [3,12,13]. However, the effect of temperature on the mechanical behavior of rubber materials in a larger deformation range (150% strain) is rarely studied. Meanwhile, due to the temperature correlation of different constitutive models having rarely been studied, the hyper-elastic behavior of rubber at different temperatures under a constant elongation strain can be accurately described by different constitutive models. This is helpful to select a fitted constitutive model that can better characterize the hyper-elastic behavior of rubber specimens at various temperatures.

This paper aims to systematically investigate the influence of temperature on the hyper-elastic mechanical behavior of filled rubbers. Section 2 introduces the commonly used hyper-elastic constitutive model for carbon black-filled rubbers. Section 3 displays the materials and the experimental setup. Section 4 shows different hyper-elastic stress–strain curves of unfilled and filled rubbers with various CB contents at different temperatures. Section 5 not only reveals the relationship between the Yeoh model parameters and the ambient temperature but extends the Yeoh model to an explicit temperature-dependent model. The evaluation results show that the model can accurately reveal the effect of temperature on the hyper-elastic behavior of tire rubber. Combining the relationship between the parameters of the Yeoh constitutive model and the ambient temperature, an explicit temperature-dependent Yeoh constitutive model was developed and has been applied to the FEA. Finally, Section 6 comes to the conclusion.

## 2. Constitutive Models

Current rubber hyper-elastic constitutive models can be divided into two main categories: one is the molecular network model based on the theory of thermodynamics statistics to examine the conformational entropy change of the molecular network. These models can predict the hyper-elastic mechanical behavior of large strains with fewer model parameters [14]. The other is the image-only models based on the continuum medium mechanics theory to simulate the elastic response of unfilled and filled rubbers at large strains [15,16].

In the following sections, we will thoroughly introduce three constitutive models, including the Yeoh model, the Ogden model, and the Arruda–Boyce model.

### 2.1. Yeoh Model

The most general strain energy density function in the form of a deformation tensor invariant series, first proposed by Rivlin [17], was taken as the reduced polynomial model and can be expressed as:(1)$$W=\sum_{i+j=1}^{N}C_{i}{}_{j}{\left(I_{1}-3\right)}^{i}(I_{2}-3{)}^{j}$$

Yeoh proposed a simplified polynomial strain energy function after analyzing the experimental data of filled rubber. The incompressible Yeoh model assumes that the strain energy function is only a general polynomial in the first principal stretching invariant. For the reduced polynomial model Equation (1) with *N* = 3, the Yeoh constitutive model is obtained and can be expressed:(2)$$W=C_{10}(I_{1}-3)+C_{20}{\left(I_{1}-3\right)}^{2}+C_{30}{\left(I_{1}-3\right)}^{3}$$

The Yeoh constitutive model can produce a typical S-shaped stress–strain relationship curve, which is more in line with the highly nonlinear mechanical properties of the hyper-elastic rubber material. It is generally believed that the Yeoh model has better accuracy in the larger deformation range of rubber, and, therefore, the Yeoh model is widely used in practical engineering analysis [18,19].

The relationship between the nominal stress f and the stretch ratio *λ* can be expressed as follows:(3)$$f=2(\lambda^{2}-\frac{1}{\lambda })\left[C_{10}+2C_{20}(\lambda^{2}+\frac{2}{\lambda }-3)+3C_{30}{\left(\lambda^{2}+\frac{2}{\lambda }-3\right)}^{2}\right]$$
where $C_{10}$, $C_{20}$, and $C_{30}$ are the parameters of the material model, which can be determined by uniaxial tensile tests. The characteristics of the Yeoh model are mainly reflected in the small strain state, which represents the initial shear modulus. $C_{10}=\mu /2$ is the Yeoh constitutive model and is half of the initial shear modulus at a small strain. $C_{20}$ is the softening phenomenon at a medium deformation, and it is generally negative. $C_{30}$ shows that the material becomes hard again within a large deformation range.

### 2.2. Ogden Model

Ogden [20] directly expressed the strain energy density function in terms of the principal elongation ratio $\lambda_{i}$, which was used as an independent variable and can be expressed as:(4)$$W=\sum_{i=1}^{N}\frac{\mu_{i}}{\alpha_{i}}(\lambda_{1}^{\alpha_{i}}+\lambda_{2}^{\alpha_{i}}+\lambda_{3}^{\alpha_{i}}-3)$$
where $\mu_{i}$ and $\alpha_{i}$ are arbitrary constants (they can be non-integers). The analytical accuracy of the Ogden model gradually increases with the increase in the polynomial order N, suggesting a relatively large flexibility of the Ogden model. However, the value of *N* is generally not greater than 4. To satisfy stability, the value of $\sum_{i}^{N}\mu_{i}\alpha_{i}$ should be greater than 0. Due to more or fewer problems with the computational accuracy of the lower-order model, it is very difficult to accurately fit a large number of intrinsic parameters of the higher-order model. It is also believed that the fourth-order or higher-order strain energy density functions are no longer of much practical value. Therefore, it is not recommended to choose a higher-order model for calculation, and the third-order Ogden constitutive model is commonly used in engineering.

The relationship between the nominal stress f and the stretch ratio λ can be expressed as:(5)$$f=\mu_{1}(\lambda^{\alpha_{1}}{}^{-1}-\lambda^{-\frac{1}{2}\alpha_{1}-1})+\mu_{2}(\lambda^{\alpha_{2}}{}^{-1}-\lambda^{-\frac{1}{2}\alpha_{2}-1})+\mu_{3}(\lambda^{\alpha_{3}}{}^{-1}-\lambda^{-\frac{1}{2}\alpha_{3}-1})$$
where $\mu_{i}$ and $\alpha_{i}$ are material constants.

### 2.3. Arruda–Boyce Model

The model based on Langevin statistical theory proposed by Arruda and Boyce (1993) is a non-Gaussian chain network model [14]. The Arruda–Boyce model is also known as the eight-chain model, which can be expanded to the form of the Taylor series (only the first five terms are retained) as:(6)$$W=C_{R}\sum_{i=1}^{5}\frac{C_{i}}{n^{i-1}}(I_{1}^{i}-3^{i})$$
where $C_{1}=\frac{1}{2},C_{2}=\frac{1}{20},C_{3}=\frac{11}{1050},C_{4}=\frac{19}{7000},C_{5}=\frac{519}{673750}$. $C_{R}=Nk\theta$ is the first strain invariant.

The relationship between the nominal stress f and the stretch ratio λ can be expressed as:(7)$$f=C_{R}(\lambda -\frac{1}{\lambda^{2}})\left[3+\frac{3}{5N}(\lambda^{2}+\frac{2}{\lambda })+\frac{33}{175N^{2}}{\left(\lambda^{2}+\frac{2}{\lambda }\right)}^{2}+\frac{57}{875N^{3}}{\left(\lambda^{2}+\frac{2}{\lambda }\right)}^{3}+\frac{1557}{67375N^{4}}{\left(\lambda^{2}+\frac{2}{\lambda }\right)}^{4}\right]$$
where $C_{R}$ and N are material constants. The parameter N is independent of temperature in the physical sense. For filled rubbers, N is temperature-dependent due to the steric dislocation effect of carbon black particles on polymer chains inside the rubber material in space [12,14,21]. In a large deformation strain range, the eight-chain model provides more accurate calculations even though there is only a small amount of material behavior known, because it has only two parameters.

## 3. Experimental Setup

### 3.1. Experimental Materials

Four rubber specimens with different contents of carbon black were used for the experiments. The rubber matrix was natural rubber, and the filled carbon black was N220. Among the four rubber formulations, only the filling amount of carbon black was different. The filler mass fractions of carbon black in the four types of rubbers, i.e., C00, C20, C40, and C60, were 0 phr, 20 phr, 40 phr, and 60 phr, respectively. The rubber types and formulations used in the test are shown in Table 1. The natural rubber was obtained from Shandong Haoshun Chemical Co., Ltd. in Jinan, Shandong, China. The carbon black N220 was obtained from Tianjin Zhengning New Material Co, Ltd. (Tianjin, China). The other stearic acid, zinc oxide, sulfur, accelerator NS, and antioxidant 4020 were all commercially available industrial-grade products.

### 3.2. Sample Preparation

The rubber compounding was divided into two stages. First, the natural rubber was pressed for 3 min on a double-roller opener (Model: S(X)-160A, Shanghai First Rubber Machinery Co., Ltd., Shanghai, China). The finished natural rubber was put into the Hakke torque rheometer (HAAKE Germany, Germany), and then the zinc oxide, stearic acid, antioxidant 4020, and carbon black were added sequentially and mixed for 10 min. The second mixing stage was conducted on a two-roller kneader (Shanghai Rubber & Plastic Machinery Co. Shanghai, China). The sulfur and accelerator NS were added to a blended section of the mixture and blended well. Then, the mixture was pressed into 3–5 mm sheets and left to stand for about 5 h before vulcanization. The vulcanization conditions were 150 °C, 10 MPa, and equivalent vulcanization time (Tc90). According to ISO 37-2017, specimens of dumbbell type 2 were prepared, and the thickness of the rubber specimen was 2 mm. The double eccentric wheel fixture RA-4-1 was used for uniaxial tensile testing, which is a special tensile fixture for rubber. The force and displacement accuracy of the universal electronic tensile tester (Taiwan High-Tech Testing Instruments Co, Taiwan, China) was 0.5, the temperature control accuracy was ±1 K, and the scale distance of the displacement sensor was 20 mm. To ensure the specimen reached the required test temperature, the temperature of the chamber needs to stabilize for 10 min after reaching the test temperature before starting the experiment. To eliminate the Mullins effect (the stress softening effect of rubber materials) [3], rubber specimens were loaded and unloaded at a rate of 100 mm/min for 10 cycles. The purpose of this step is to more accurately reproduce the working condition of the tire. The modulation strain should be the 150% strain, and the modulation temperature should be set to 288 K. After modulation, the rubber specimens should stand for more than 24 h to fully recover the elastic deformation and exhibit stable properties. The experiments were repeated at least five times under each condition, and the average value was taken as the final experimental result.

### 3.3. Test Results

The deformation of rubber components in engineering applications is generally less than 100%. In some extreme operating conditions, such as the rolling of the tire on a raised road surface, the stretching of the rubber spring may be greater than the 100% strain. Therefore, the 150% strain was used to characterize the mechanical properties of the rubber material in the uniaxial tensile test. Figure 1 shows the relationship between the nominal stress and nominal strain for the C00, C20, C40, and C60 rubber specimens in Table 1. The experimental temperatures were 293 K, 313 K, 333 K, 353 K, 363 K, and 383 K, respectively. Figure 1(a2–d2) locally magnify the stress–strain differences between the four rubber specimens at different temperatures. From the strain–strain curve in Figure 1, it can be observed that the hyper-elastic mechanical behavior of the rubber material shows a more pronounced temperature dependence over a deformation range of 150%. For the carbon-black-filled rubber specimens C20, C40, and C60, the rubber samples first became “soft” with increasing temperature, and when the temperature reached a certain temperature, the rubber samples gradually became “hard” as the temperature rose. The turning temperature was different for different rubber samples. Therefore, the temperature dependence of the hyper-elastic mechanical behavior of rubber materials can be considered the result of two mechanisms. One is the “positive effect” that hardens the rubber sample, and the other is the “negative effect” that softens the rubber sample. For natural rubber C00 unfilled with carbon black, the stress–strain curve always increased with increasing temperatures. There was no change from soft to hard. The stress–strain curve does not show a clear turning temperature.

## 4. Discussion

From the uniaxial tensile experimental data at different temperatures, a preliminary study on the temperature dependence of rubber between the Yeoh model, Ogden model, and Arruda–Boyce model was carried out on the C20 rubber specimen. The smaller the residual sum of squares (*RSS*), the closer the fit is to 1. In order to obtain the fitting ability of the hyper-elastic intrinsic model more quickly, the residual sum of squares (*RSS*) was calculated to evaluate the fitting ability of the Arruda–Boyce model, Ogden model, and Yeoh model.
(8)$$RSS=\sum_{i=1}^{N}(P_{i}-{\hat{P}}_{i}{)}^{2},TSS=\sum_{i=1}^{N}(P_{i}-\bar{P}{)}^{2},R^{2}=1-RSS/TSS$$
where $P_{i}$ is the experimental value; $\bar{P}$ is the average of the test values; ${\hat{P}}_{i}$ is the model fit value; *N* is the number of experimental data points involved in the fit. The smaller the *RSS*, the larger the *R*^2^, indicating a better overall fit of the model.

From Figure 2 and Figure 3, it can be seen that the Arruda–Boyce model shows general “S-shaped” stress–strain characteristics of the hyperelastic behavior of carbon black-filled rubber at different temperatures. However, there are still some obvious deviations between the fitted results and the experimental data. It was difficult for the Arruda–Boyce model to reflect the nonlinear characteristics of the hyper-elastic mechanical behavior of carbon black-filled rubber at different ambient temperatures under the 150% strain. This was consistent with the conclusion summarized in the previous theoretical presentation. Meanwhile, the Arruda–Boyce model cannot reflect the nonlinear characteristics of the hyper-elastic mechanical behavior of carbon black-filled rubbers in small and medium deformations well, showing a more significant error with experimental data. The Ogden model (*N* = 3) and Yeoh model also show an “S-shaped” stress–strain curve for the hyper-elastic behavior of carbon black-filled rubber at different temperatures. This indicates that the fitted curves of the two models can reasonably describe the experimental data under the 150% strain.

Table 2 lists the parameters fitting of the Ogden constitutive model (*N* = 3) with the experimental data of the C20 rubber specimen at different temperatures. Figure 4 shows the trend of the parameters of the uniaxial tensile data with temperature. The Ogden constitutive model (*N* = 3) fit the experimental data in Figure 2 and Figure 3 well. Figure 4 and Table 2 show the relationship between the parameters of the Ogden model (*N* = 3) and temperature. As can be seen from the diagram, the parameters of the Ogden model have no law with the change of temperature, so it can be said that the parameters have no temperature dependence. Due to the excessive parameters of the Ogden model (*N* = 3), the model did not converge easily when fitting to the experimental data, resulting in longer computation times and a low applicability of the model.

From Table 3 and Figure 5, it can be seen that the material parameters $C_{10},C_{20}$, and $C_{30}$ vary approximately as a quadratic function with temperature, which indicates that the material parameters are correlated with temperature. As the temperature gradually increased, the rubber gradually softened, and the shear modulus decreased. The reason for this is that the material parameter $C_{10}$ indicates the initial shear modulus at small strains. However, the rubber started to “harden” after reaching the turning temperature. The ability of the carbon black-filled rubber to resist the strain increased, and the shear modulus gradually rose. This was consistent with the trend of the experimental data in Figure 1(b1). The material parameter $C_{20}$ indicates the softening phenomenon of the filled rubber in a medium deformation. The larger the material parameter $C_{20}$, the more obvious the softening phenomenon, and when it came to the turning temperature, the filled rubber was the softest, and the material parameter $C_{20}$ was the largest. The material parameter $C_{30}$ indicates the phenomenon that the material started to harden again in a large deformation, and after reaching the turning temperature, the material parameter $C_{30}$ became larger due to the hardening of the filled rubber. Therefore, there was significant dependence between the Yeoh model and temperature, and this temperature dependence can be expressed by numerical fitting using the quadratic function.

In summary, the ArrudaBoyce model cannot reflect the nonlinear characteristics of the hyper-elastic mechanical behavior of carbon black-filled rubbers in small and medium deformations well, showing a more significant error with experimental data. Although the Ogden constitutive model (*N* = 3) fit the experimental data well, there also existed an irregularity of the model parameters with temperature. Due to the excessive parameters of the Ogden model (*N* = 3), the model did not converge easily when fitted to the experimental data, resulting in longer computation times and a low applicability of the model. On that basis, it can be concluded that there was significant dependence between the Yeoh model and temperature, and this temperature dependence can be expressed by numerical fitting using the quadratic function.

Using the Yeoh constitutive model, the relationship between the material parameters and temperature can be expressed as:(9)$$\left\{ \begin{array}{l} C_{10}=A_{0}+A_{1}T+A_{2}T^{2}\\ C_{20}=B_{0}+B_{1}T+B_{2}T^{2}\\ C_{30}=C_{0}+C_{1}T+C_{2}T^{2} \end{array}\right.$$
where $A_{0},A_{1},A_{2},B_{0},B_{1},B_{2},C_{0},C_{1},C_{2}$ are the temperature-dependent parameters of the Yeoh constitutive model, which can be determined by fitting the Yeoh constitutive model. For different volume fractions of carbon black-filled rubbers, the temperature-dependent parameter values can be obtained by fitting the parameters of the Yeoh constitutive model at different temperatures by Equation (9). The details are shown in Table 4, Table 5 and Table 6 and Figure 6.

By combining Equations (2) and (9), the Yeoh constitutive model with the explicit temperature parameter can be obtained.
(10)$$\left\{ \begin{array}{l} W=C_{10}(I_{1}-3)+C_{20}{\left(I_{1}-3\right)}^{2}+C_{30}{\left(I_{1}-3\right)}^{3}\\ C_{10}=A_{0}+A_{1}T+A_{2}T^{2}\\ C_{20}=B_{0}+B_{1}T+B_{2}T^{2}\\ C_{30}=C_{0}+C_{1}T+C_{2}T^{2} \end{array}\right.$$

Based on Equation (10) and the parameters in Table 4, Table 5 and Table 6, the stress–strain curves for four different contents of carbon black-filled rubbers at different temperatures were plotted, which can be used to predict the trend of the parameters of the Yeoh constitutive model with explicit temperature parameters. Figure 7 shows the prediction curves of the model (a1–d1) with local zoom-in plots (a2–d2). From Figure 7, the predicted results of the Yeoh constitutive model with the explicit temperature parameter are in good agreement with the experimental results. This indicates that the Yeoh constitutive model with the explicit temperature parameter can more accurately describe the nonlinear hyper-elastic mechanical behavior of carbon black-filled rubber at different temperatures under the 150% strain.

To visualize the relationship between the effect of temperature on the “softness” and “hardness” of the rubber, the correlation between the adhesive stress and the temperature at different constant elongation strains of 0.2, 0.6, 1, and 1.4 was investigated. From Figure 8, it can be seen that, as the temperature increased, the stress in the rubber specimens (C20, C40, C60) at a constant elongation first decreased with the increase in temperature and then, after reaching the turning temperature, increased again with the increase in temperature. The carbon black-filled rubber sample first became “soft” with the increase in temperature and then gradually became “hard” when it reached the turning temperature. The temperature changed with the number of carbon black-filled masses. The stress transition temperature increased with the increase in the volume fraction of the carbon black. However, for the unfilled carbon black rubber specimen C00, the stress tended to increase roughly linearly with the increasing temperature at different constant elongation strains. The above results were the same as the conclusion of the stress–strain curves measured by the above tests.

There are two reasons for the temperature dependence of the hyper-elasticity of carbon black-filled rubbers. Firstly, due to the gradual increase in temperature, the movement between molecules is more intense. The intermolecular potential energy is reduced, which results in a thermal softening effect of the filled rubber. Secondly, due to the gradual increase in temperature, the conformational entropy of the long-chain molecular system of rubber changes. The thermos-elasticity of the rubber is enhanced, which makes the filled rubber show a thermal hardening effect. The thermal softening effect of the filled rubber plays a major role at lower temperatures. When the test temperature exceeds the turning temperature, the thermal hardening effect of the filled rubber gradually plays a major role. Therefore, the phenomenon of “softening first, then hardening” of the filled rubber occurs with the increase in temperature. Meanwhile, due to the addition of carbon black, the thermal softening effect of the filled rubber increases with the increase in the volume fraction of the carbon black filling, the thermal hardening effect gradually decreases, and the turning temperature becomes higher and higher.

## 5. Application of the Yeoh Model with Explicit Temperature Parameters in FEA

From Equation (10) and Table 4, Table 5 and Table 6, the temperature-dependent characterization parameters of rubber specimens with four different carbon black-filled mass fractions can be obtained from the model parameters at different temperatures. Then, uniaxial tensile simulations were performed on C60 rubber specimens at 293 K using ABAQUS/CAE. The simulation results were compared with the experimental data. A dumbbell-shaped model with the same properties as the experiment was built using ABAQUS/CAE. Then, the uniaxial tensile model was obtained by adding material properties, building components, setting analysis steps, and dividing meshes, as shown in Figure 9. The tensile specimen model used the C3D8RH unit. In order to resemble the experimental process as much as possible, this simulation coupled the specimen area with reference points A and B. In addition, the boundary condition that A is completely fixed and the displacement along the Y-axis is applied to the reference point B was set. The tensile process of the universal power tensile tester was simulated.

It can be seen from Figure 10 that the stress distribution in the middle part of the dumbbell model is more uniform, and the stress concentration region of the specimen is a circular arc from narrow to wide. The stress–strain curve for the simulation was also determined from the uniform deformation region in the middle part of the dumbbell model. From Figure 11, the simulated uniaxial tensile data are more consistent with the trend of the experimental data. This indicates that the Yeoh constitutive model with apparently included temperature parameters can predict the uniaxial tensile test data at different temperature ranges under the 150% strain. Because the model parameters can be obtained through uniaxial tensile tests, they can be better applied to actual working conditions, with a guarantee of certain accuracy requirements. Therefore, the Yeoh constitutive model with explicit temperature parameters has high engineering applicability.

If the temperature of the rubber specimen is known, the corresponding model parameters can be calculated immediately from the Yeoh constitutive model with explicit temperature parameters. Therefore, the Yeoh constitutive model with explicit temperature parameters can be quickly applied to finite element analysis. The Yeoh constitutive model with explicit temperature parameters provides a more convenient and accurate method for the analysis of other hyper-elastic finite element models. However, the simulation results of the Yeoh constitutive model with explicit temperature parameters still have some deviations from the experimental data. This indicates that there is still room for improvement in the model.

## 6. Conclusions

Based on the Yeoh constitutive model and continuum medium mechanics theory, the Yeoh constitutive model with explicit temperature parameters was constructed. According to the uniaxial tensile experimental data of rubber samples, the following conclusions can be obtained:(1)The hyper-elastic mechanical behavior of the carbon black-filled rubber specimens was strongly correlated with temperature in a large deformation range (150% strain). The turning temperature was correlated with the carbon black-filled mass fraction, which gradually increased with the increase in the carbon black-filled mass fractions. The turning temperatures were 333 K, 353 K, and 363 K. For the unfilled carbon black rubber, its stress–strain curve always increased with the increase in temperature.(2)The Yeoh model, Ogden model, and Arruda–Boyce model were investigated to characterize the hyper-elastic mechanical behavior of rubbers at different temperatures. Although the Ogden constitutive model fits the stress–strain curves at different temperatures well, there was no regularity in the variation of its model parameters with temperature. The Yeoh constitutive model can fit the stress–strain curves at different temperatures well with the experimental results, and it can exhibit temperature-dependent hyper-elastic mechanical behavior in a wide range of deformations. Based on the Yeoh constitutive model and the continuous medium mechanics theory, the Yeoh constitutive model with explicit temperature parameters was constructed, which can better describe the constitutive mechanical behavior of rubber at different temperatures.(3)The finite element analysis of uniaxial stretching was performed using the Yeoh constitutive model with explicit temperature parameters. The simulation results were in general agreement with the experimental data, and the rationality of the model was verified.

## Figures and Tables

**Figure 1 polymers-14-03521-f001:**
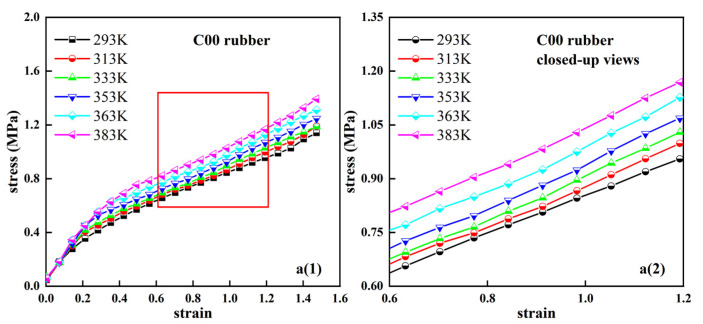
Stress–strain fitting curves of four types of rubber compounds at different temperatures based on the eight-chain model. (**a1**,**b1**,**c1**,**d1**): fitting curves and experimental data of four types of rubbers; (**a2**,**b2**,**c2**,**d2**): corresponding local closed-up views.

**Figure 2 polymers-14-03521-f002:**
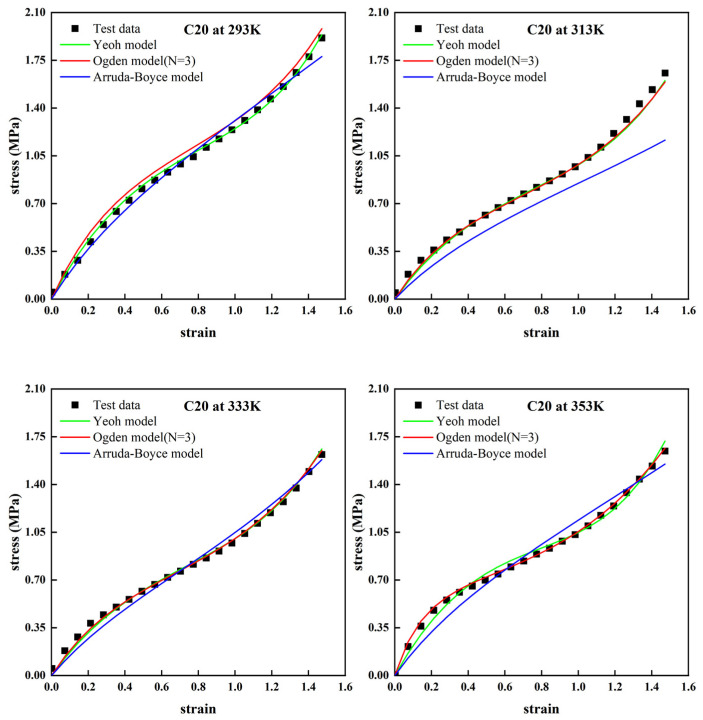
Fitting results of the Yeoh model, Ogden model (*N* = 3), and Arruda–Boyce model for unidirectional stretching of the C20 rubber specimen at different temperatures.

**Figure 3 polymers-14-03521-f003:**
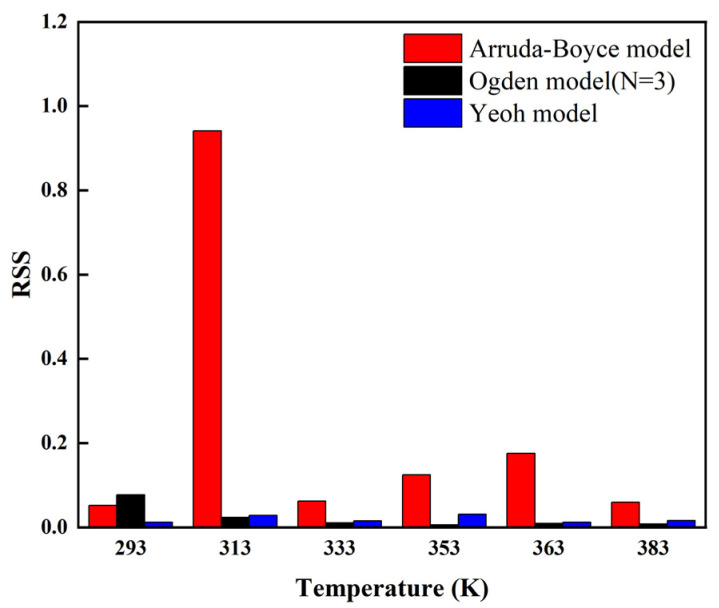
Residual sums of squares of the fit of the Yeoh model, Ogden model (*N* = 3), and Arruda–Boyce model for uniaxial stretching of the C20 rubber specimen.

**Figure 4 polymers-14-03521-f004:**
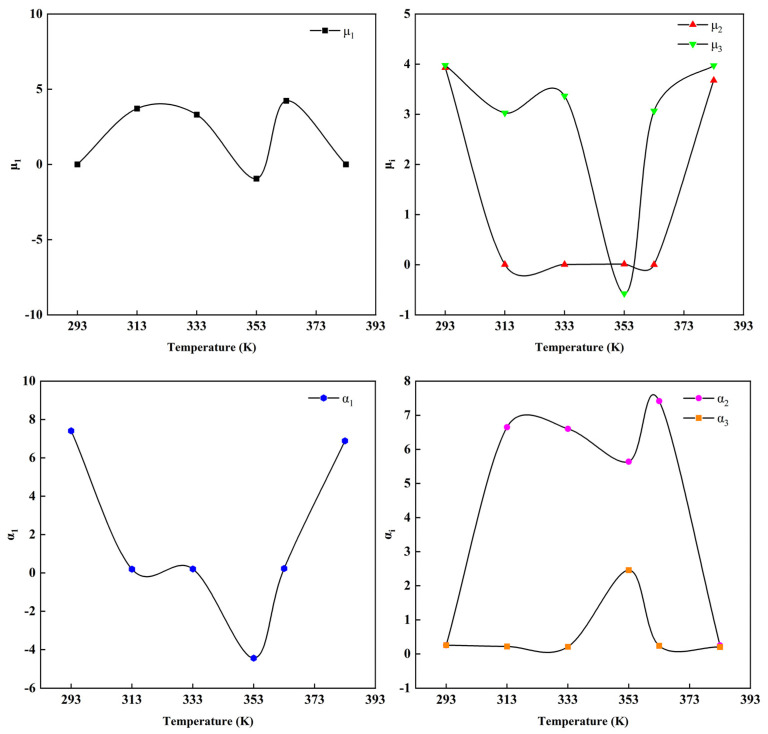
Trend of parameters for the uniaxial tensile data of the C20 rubber specimen at different temperatures by the Ogden constitutive model (*N* = 3).

**Figure 5 polymers-14-03521-f005:**
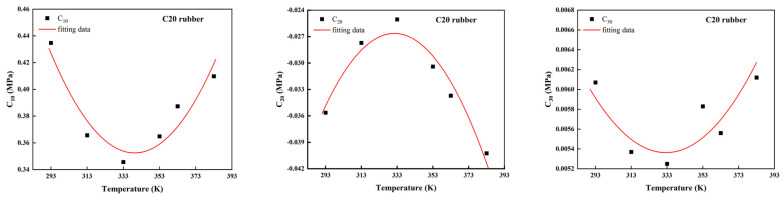
Trends of parameters for the uniaxial tensile data of the C20 rubber specimen at different temperatures by the Yeoh model.

**Figure 6 polymers-14-03521-f006:**
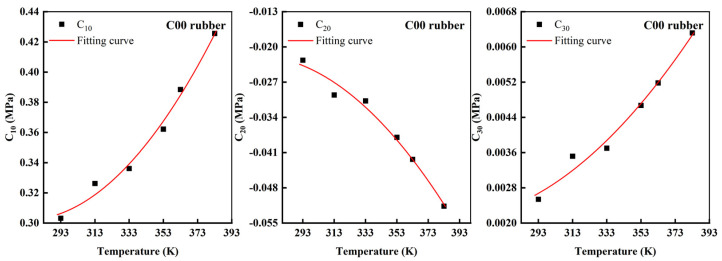
Temperature—dependent quadratic function fitting curves of parameters of the Yeoh constitutive model for four types of rubbers.

**Figure 7 polymers-14-03521-f007:**
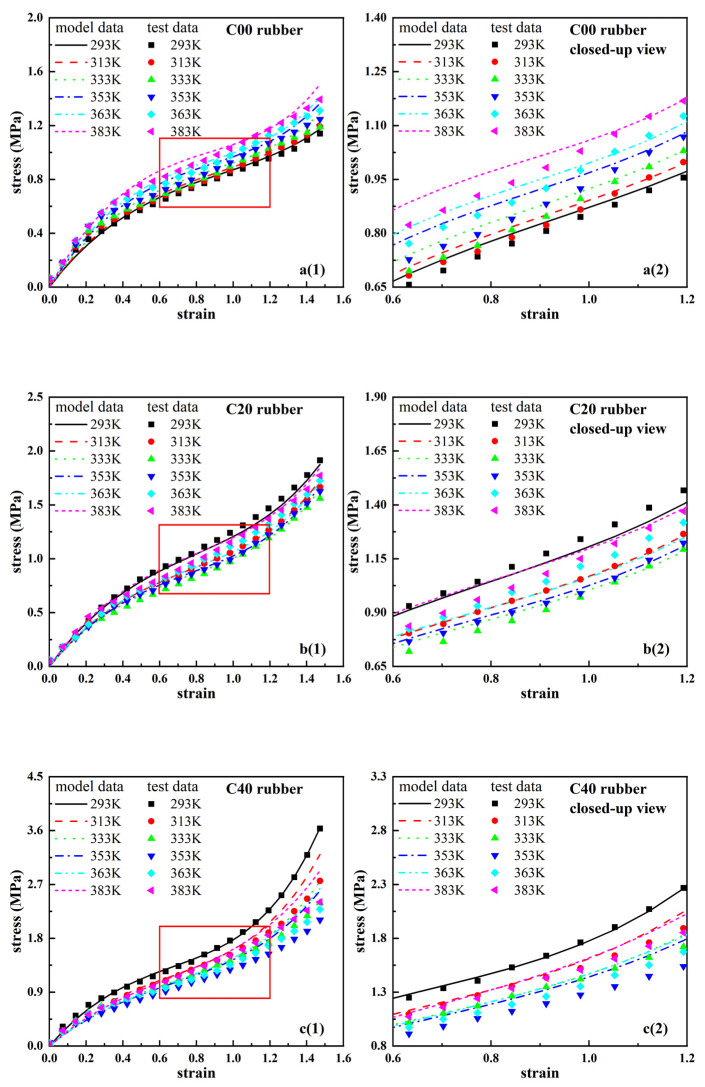
Model stress–strain curves of four types of rubbers at different temperatures based on the Yeoh constitutive model with the explicit temperature parameter. (**a1**,**b1**,**c1**,**d1**): model curves and experimental data of four types of rubbers; (**a2**,**b2**,**c2**,**d2**): corresponding local closed-up views.

**Figure 8 polymers-14-03521-f008:**
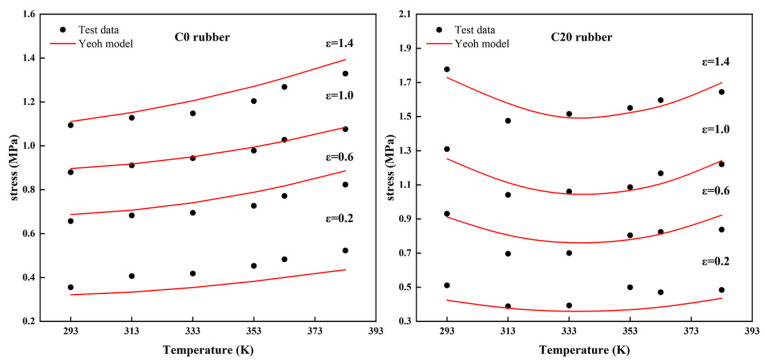
Stress–temperature curves of four types of rubber specimens at different constant elongation strains and the prediction curves of the Yeoh constitutive model with explicit temperature parameters.

**Figure 9 polymers-14-03521-f009:**
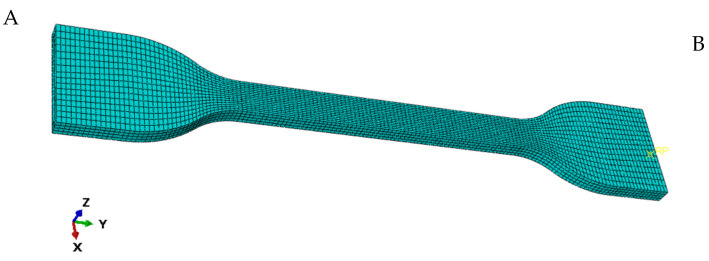
Uniaxial tensile specimen model and meshing.

**Figure 10 polymers-14-03521-f010:**
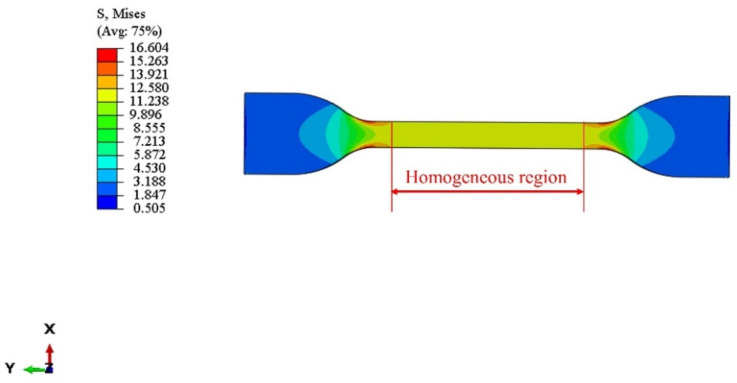
Contour map of the stress of the finite element analysis (FEA) of the C60 rubber specimen at 293 K.

**Figure 11 polymers-14-03521-f011:**
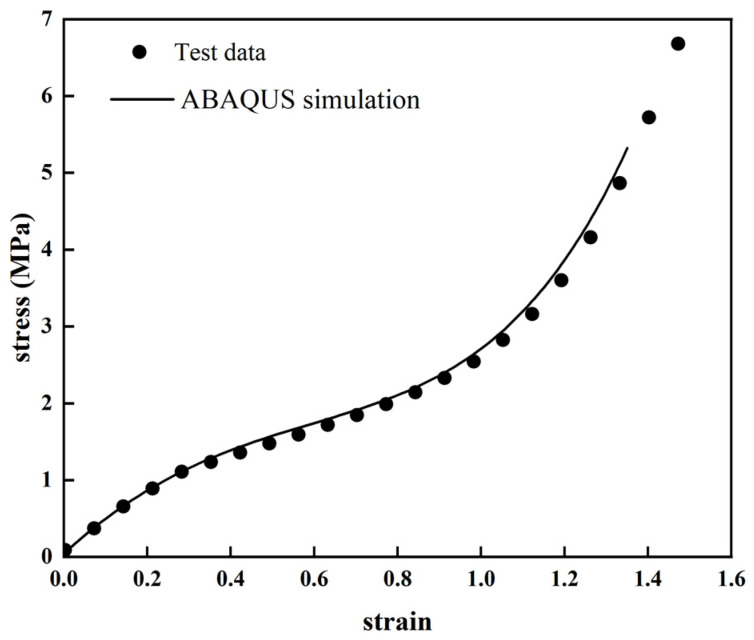
The stress–strain curve of the FEA results and the experimental data of the C60 rubber specimen.

**Table 1 polymers-14-03521-t001:** Formulas and codes of four kinds of carbon black-filled vulcanized rubber (UNIT: PHR).

Code	C00	C20	C40	C60
NR	100	100	100	100
CB N220	0	20	40	60
Zinc oxide	5	5	5	5
Stearic acid	3	3	3	3
Sulfur	2	2	2	2
Accelerator NS	1	1	1	1
Antioxidant 4020	1.2	1.2	1.2	1.2
Total	112.2	132.2	152.2	172.2

**Table 2 polymers-14-03521-t002:** Parameters fitting of the Ogden constitutive model (*N* = 3) with the experimental data of the C20 rubber specimen at different temperatures.

Temperature (K)	$\mu_{1}$	$\mu_{2}$	$\mu_{3}$	$\alpha_{1}$	$\alpha_{2}$	$\alpha_{3}$	RSS
293	0.002418	3.93	3.977	7.409	0.2559	0.2554	0.07695
313	3.713	0.004769	3.028	0.1954	6.648	0.2178	0.02325
333	3.311	0.005351	3.367	0.2061	6.601	0.2071	0.01071
353	−0.9414	0.0135	−0.5741	−4.441	5.637	2.455	0.00599
363	4.231	0.002228	3.067	0.2317	7.414	0.2382	0.00955
383	0.00379	3.678	3.973	6.887	0.2494	0.2022	0.00826

*RSS*—Residual Sum of Squares.

**Table 3 polymers-14-03521-t003:** Parameters fitting of the Yeoh model with the experimental data of the C20 rubber specimen at different temperatures.

Temperature (K)	$C_{10}$	$C_{20}$	$C_{30}$	RSS
293	0.43468	−0.03564	0.00607	0.01226
313	0.36566	−0.02772	0.00537	0.02866
333	0.34564	−0.02506	0.00525	0.01571
353	0.36487	−0.03041	0.00583	0.03074
363	0.38733	−0.03371	0.00556	0.01225
383	0.40977	−0.04025	0.00612	0.01588

**Table 4 polymers-14-03521-t004:** Temperature-dependent characterization parameters of the model parameter $C_{10}$ for four types of rubbers.

Code		*C* _10_		
	*A* _0_	*A* _1_	*A* _2_	*R* ^2^
C00	1.04371	−0.00545	1.00 × 10^−5^	0.99985
C20	4.36529	−0.02366	3.49 × 10^−5^	0.99933
C40	5.76015	−0.03	4.22 × 10^−5^	0.99896
C60	4.70282	−0.0219	2.97 × 10^−5^	0.99781

**Table 5 polymers-14-03521-t005:** Temperature-dependent characterization parameters of the model parameter $C_{20}$ for four types of rubbers.

Code		*C* _20_		
	*B* _0_	*B* _1_	*B* _2_	*R* ^2^
C00	−0.15676	0.00104	−1.99 × 10^−6^	0.99886
C20	−0.46465	2.64 × 10^−3^	−4.00 × 10^−6^	0.97783
C40	−0.83582	0.00428	−5.64 × 10^−6^	0.97192
C60	−0.45638	0.00125	−4.50 × 10^−7^	0.96902

**Table 6 polymers-14-03521-t006:** Temperature-dependent characterization parameters of the model parameter $C_{30}$ for four types of rubbers.

Code		*C* _30_		
	*C* _0_	*C* _1_	*C* _2_	*R* ^2^
C00	0.01323	−9.42 × 10^−5^	1.98 × 10^−7^	0.9987
C20	0.04464	−2.36 × 10^−4^	3.56 × 10^−7^	0.9952
C40	0.24883	−0.00132	1.80 × 10^−6^	0.98163
C60	0.41466	−0.00205	2.59 × 10^−6^	0.98886

*R*^2^—Coefficient of determination

## Data Availability

The data presented in this study are available on request from the corresponding author.

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
