# Peer review of "Temperature Dependence of Rubber Hyper-Elasticity Based on Different Constitutive Models and Their Prediction Ability"

_polymers, 2022, doi:10.3390/polym14173521_

Round 1

Reviewer 1 Report

In this paper, they develop a method to evaluate the hyperelastic behavior of rubber at different temperatures. The developed way is well explained and can be expanded to different polymers, making it useful in different research areas. 

I just have some comments that I think can make the manuscript clear:

1. As a general comment, please remove the material and method explanation in part 2. consecutive models. These are guidelines for the authors and don´t need to be in the published draft. 

2. The authors can make a summary of the key differences between the models they use just before they start developing their maths to make more clear for the reader what we are expecting. 

Author Response

We would like to thank you for your careful reading, helpful comments, and constructive suggestions, which has significantly improved the presentation of our manuscript.

We have carefully considered all comments from the reviewers and revised our manuscript accordingly. The manuscript has also been double-checked, and the typos and grammar errors we found have been corrected. 

Author Response

(The authors gave the same response as above.)

Round 2

Reviewer 2 Report

The paper can be published in the corrected version.

The reviewer suggest to add post-mortem MEB images in order to understand the effect of the carbon black.